# Effect of Microencapsulated Medium-Chain Fatty Acids, Lignocellulose, and Heat-Killed *Lactobacillus plantarum* L-137 Supplementation on Lactating Sow Performance, and Nutritional and Immunological Parameters in Colostrum

**DOI:** 10.3390/vetsci12020134

**Published:** 2025-02-06

**Authors:** Nithat Wichasit, Wandee Tartrakoon, Rangsun Charoensook, Riantong Singanusong, Juan J. Loor, Gaku Shoji, Satoru Onoda, Tossaporn Incharoen

**Affiliations:** 1Department of Agricultural Science, Division of Animal Science and Feed Technology, Faculty of Agriculture, Natural Resources and Environment, Naresuan University, Phitsanulok 65000, Thailand; nithatw61@nu.ac.th (N.W.); rangsunc@nu.ac.th (R.C.); 2Center of Excellence in Nonlinear Analysis and Optimization, Faculty of Science, Naresuan University, Phitsanulok 65000, Thailand; 3Department of Ago-Industry, Faculty of Agriculture, Natural Resources and Environment, Naresuan University, Phitsanulok 65000, Thailand; riantongs@nu.ac.th; 4Centre of Excellence in Fats and Oils, Faculty of Agriculture, Natural Resources and Environment, Naresuan University, Phitsanulok 65000, Thailand; 5Department of Animal Sciences, Division of Nutritional Sciences, University of Illinois, Urbana, IL 61801, USA; jloor@illinois.edu; 6House Wellness Foods Corporation, 3-20 Imoji, Itami 664-0011, Japan; shoji@housefoods.co.jp (G.S.); onoda_satoru@house-wf.co.jp (S.O.)

**Keywords:** medium-chain fatty acids, lignocellulose, heat-killed *Lactobacillus plantarum* L137, sows, suckling pig, colostrum quality

## Abstract

Feeding and nutrition of sows during late-gestation and lactation are crucial for maximizing piglet production and survival. In this study, the administration of microencapsulated medium-chain fatty acids (miMCFA) demonstrated significant enhancement of live-born piglet numbers, and elevated colostrum yield. Furthermore, the synergistic application of miMCFA in conjunction with lignocellulose and heat-killed *Lactobacillus plantarum* L137 (HKL137) exhibited additional neonatal piglet benefits and sow performance, specifically in terms of fat content and immunoglobulin G and M (IgG and IgM) concentrations in colostrum. Compared to the control group, sows with enhanced immune transfer produced 60.5% more colostrum during their first piglet parturition period. The colostrum exhibited increased concentrations of lipids (28%) along with significant IgG and IgM levels. These results highlight our strategic feeding leads to higher colostrum quality resulting in increasing the number of live-born piglets.

## 1. Introduction

Feeding and nutrition of sows during late-gestation and lactation are crucial for maximizing piglet production and survival. During this stage, sows face increased nutritional demands as they support both their own needs and the development of their offspring. Adequate nutrition during this period has been associated with enhanced birth weights, higher survival rates, and improved overall health of piglets, which ultimately impacts the productivity of the swine industry [1]. Among the various nutritional components, fatty acids play a pivotal role. They are not only a primary energy source but also contribute to various physiological functions including metabolic regulation, antibacterial activity, and anti-inflammatory responses [2,3,4].

Previous studies have shown similar improvements in colostrum composition and piglet growth following supplementation with fatty acid supplementation including β-hydroxy β-methyl butyrate, polyunsaturated fatty acids, and medium-chain fatty acids [5,6,7]. However, medium-chain fatty acids (MCFAs) are particularly important due to their unique chemical structure, characterized by fatty acids with 6 to 12 carbon atoms. These MCFAs occur naturally in milk fat and various feed materials in the form of medium-chain triglycerides (MCTs). Coconut oil and palm kernel oil are prominent sources of MCTs [8,9]. The hydrolysis of MCTs is faster than long-chain triglycerides partly due to their enhanced water solubility regardless of emulsifiers such as bile salts. MCT oils and fatty acids are efficiently absorbed and digested, making them viable energy sources for sow diets [10,11,12]. Recent studies indicated that supplementing sows with 10% coconut oil, a rich source of MCFAs, from day 84 of gestation until farrowing can significantly increase the survival rates of newborn piglets [13]. However, the challenge of rancidity in high oil content feeds necessitates alternative delivery methods. The use of oil in powdered form, through microencapsulation, offers a promising solution. Microencapsulation provides a protective barrier around the core material, allowing for the delayed release and better preservation of bioactive compounds until they reach the desired site of action in the animal’s body [14]. The encapsulation technology has directly delivered numerous bioactive substances (vitamins, organic acids, essential oils) to the intestines of animals, thus improving their efficiency [15].

Recently, dietary fiber (DF) has played a crucial role in sow nutrition. It is a carbohydrate polymer that resists digestion in the small intestine and undergoes partial or complete fermentation in the large intestine. This fermentation process enhances gut microbiota profiles, improves barrier function, and boosts immunity, particularly in weaning and growing pigs [16]. Some studies have focused on the supplementation of fiber sources such as lignocellulose into the diets of lactating sows. These studies have demonstrated that the inclusion of fiber can affect positively performance and colostrum [17] by promoting digestion, weight management, and gut microbiota health [18]. Most of the research on DF has focused on its impact on sow welfare, colostrum production, physiology, and performance [19,20]. However, there is limited information on how DF influences the quantity and quality of colostrum and milk, as well as the immune status of the offspring. Because colostrum is the primary source of maternal antibodies for newborn piglets, understanding the impact of DF on colostrum composition is critical for enhancing piglet health and growth outcomes.

Another promising area of research involves the use of heat-killed *Lactobacillus plantarum* strain L137 (HKL137), a non-viable lactic acid bacterium isolated from fermented foods. HKL137 is known for its remarkable resistance to high temperature and pressure, which makes it a stable supplement in animal feed [20,21]. HKL137 has been suggested as a candidate for use as an immunobiotic. It also plays a significant part in innate immune responses, resistance to disease and stress, and stimulation of growth in a wide variety of mammalian and aquatic animals such as mice [22] pigs [23], kuruma shrimp [24], and red sea bream [25,26]. In pigs, HKL137 has shown potential in enhancing immune responses and promoting growth, yet research on its effects in sows and their piglets remains scarce [23]. Tartrakoon et al. [27] suggested that incorporating HKL137 in pig diets can enhance production performance and boost the immune system without the inconsistencies associated with traditional in-feed probiotics. Given its immunomodulatory properties, HKL137 supplementation during late-gestation and lactation could offer significant benefits, potentially enhancing sow health, improving colostrum quality, and boosting piglet immunity. On the other hand, research on sows and piglets remains exceptionally limited.

In the current research, we aimed to fill the existing knowledge gaps by providing insights into how these dietary components interact to influence sow and piglet health, potentially leading to improved outcomes in swine production. Thus, the objective of this study was to explore the effects of supplementing sow diets with microencapsulated medium-chain fatty acids, lignocellulose, and HKL137 on lactation performance and nutritional composition and immunity in colostrum. We hypothesized that the combination of these dietary supplements would synergistically enhance sow lactation performance by improving colostrum yield and composition, resulting in improved piglet health status.

## 2. Materials and Methods

### 2.1. Feed Supplement Information

Microencapsulated medium-chain fatty acids (miMCFA) powder was produced from a mixture of crude palm kernel and soybean oils at optimized ratios. Maltodextrin and sodium caseinate served as encapsulating agents following a modified protocol based on [28]. Finally, the microencapsulation of MCFA within the wall material was achieved through a continuous spray-drying process. The main nutrient composition per 100 g of miMCFA was: protein 9.69 g, fat 31.2 g, and energy 537.96 kcal. The fatty acids in miMCFA included caprylic acid (C8:0) 1.76 g, capric acid (C10:0) 1.70 g, lauric acid (C12:0) 23.24 g, myristic acid (C14:0) 8.59 g, palmitic acid (C16:0) 7.15 g, stearic acid (C18:0) 2.02 g, oleic acid (C18:1n9t) 12.31 g, linoleic acid (C18:2n6c) 9.94 g, and α-Linolenic acid (C18:3n3) 0.03 g.

Lignocellulose (Opticell, Agromed Austria GmbH, Kremsmünster, Austria) is a feed additive primarily composed of a natural fiber complex derived from plant cell walls. It contains 30–50% cellulose, 20–30% hemicellulose, and 15–25% lignin.

HKL137 (Feed LP20, House Wellness Foods, Itami, Japan) contains 20% HKL137, and the remaining 80% dextrin on a dry-weight basis with a concentration of approximately 1.0 × 1011 cells g^−1^, were used. The product was stored at room temperature (25 ± 2 °C) until use.

### 2.2. Animal Ethics

The current investigation was conducted at the swine research farm of the Faculty of Agriculture, Natural Resources, and Environment, Naresuan University (Phitsanulok province, Thailand). Animal procedures were approved by the Naresuan University Agricultural Animal Care and Use Committee (Approval ID 6301001), in accordance with the Ethics of Animal Experimentation of the National Research Council of Thailand.

### 2.3. Experimental Design

A total of 50 multiparous Large White × Landrace sows (average parity 3–4) were assigned to five experimental groups in a completely randomized block design based on parity, body condition score (BC), backfat thickness (BF), and body weight on day 100 of gestation. The control group (CON) was fed a basal diet without supplement, while other four treatments received the basal diet with different supplements as follows: 50 g of miMCFA (S1 group), 50 g of miMCFA + 30 g of lignocellulose (S2 group), 50 g of miMCFA + 0.10 g of HKL137 (S3 group), and 50 g of miMCFA + 30 g of lignocellulose + 0.10 g of HKL137 (S4 group). Each supplemental group was administered daily in the morning meal as a topping. The experimental period commenced on day 100 of gestation and continued until seven days post-farrowing, spanning a total of 21 days.

### 2.4. Animals, Diets, and Management

The experiment was conducted at a commercial pig farm (Vilaipron Pig Farm, Nakhon Sawan Province, Thailand). The sows were maintained under a standard evaporative cooling system and housed in conventional farrowing crates measuring 2.0 × 2.5 m^2^. The environmental temperature was controlled and maintained at 25.6 ± 1.5 °C throughout the experimental period. From day 100 of gestation until farrowing, the sows in the experiment were daily fed with 3.5 kg of a standard diet. Following farrowing, the lactating sows were given ad libitum access to feed, offered three times daily at 07:00 A.M., 01:00 P.M., and 05:00 P.M. The feed consisted of a corn–soybean-based diet, formulated to meet or exceed the nutritional requirements for lactating sows according to the National Research Council (NRC) [29]. The analyzed chemical composition of the basal diet is presented in Table 1.

### 2.5. Sow Reproductive and Litter Performances

During the experiment, several sow performance parameters were recorded, including the duration of parturition, total feed intake, average daily feed intake, body weight, total piglets born, live-born piglets, survival percentage, piglet birth weight, stillborn piglets, colostrum production, and colostrum intake per piglet. Backfat thickness was measured at farrowing and again on day 7 of lactation using an ultrasonic detection device (Leanmeater^®^, Series 12, Renco Corp., Golden Valley, MN, USA) at the P2 point, which is located 60 mm to the left of the dorsal midline at the last rib. The average backfat thickness for each sow was calculated from three separate measurements, recorded in millimeters. The change in backfat thickness was determined by subtracting the measurement on day 7 of lactation from the value recorded at farrowing. Litter sizes were standardized to 11 piglets per sow 24 h post-farrowing. Also, diarrhea scores were noted individually based on the appearance of piglet feces, categorized as normal (score = 0), soft (score = 1), or runny/watery (score = 2), and assessed throughout the lactation period [30].

### 2.6. Determination of Colostrum Intake and Yield

Colostrum intake (CI) for each piglet was estimated using a method that considers the piglet’s birth weight and its weight gain during the first 24 h postpartum. This approach has been widely used in research to approximate individual piglet colostrum consumption by factoring in early energy requirements and colostrum transfer efficiency. Colostrum yield (CY) for each sow was determined as the total amount of colostrum consumed by all piglets in the litter within the first 24 h after birth. This formula, as outlined by Theil et al. [31] is expressed as: Colostrum consumption (g) = −106 + 2.26WG + 200BWB + 0.111D − 1414WG/D + 0.0182WG/BWB. Here, WG represents the piglet’s weight gain over 24 h (g), BWB stands for birth weight (kg), and D indicates the duration of colostrum suckling (min).

### 2.7. Collection of Colostrum and Milk Samples

Colostrum was manually collected from all functional teats within the first piglet parturition period, and mature milk was gathered from the same teats on day 7 of lactation [32]. To aid in milk let-down for the mature milk collection, sows were given an intravenous injection of 0.2 mL oxytocin (10 IU/mL, VetOne^®^, Boise, ID, USA). Prior to both colostrum and milk collection, the udders were cleaned with sterile water and dried with a towel to minimize contamination. About 30 mL of colostrum and mature milk were collected from all functional mammary glands into centrifuge tubes. The samples were filtered through gauze, transferred into clean 30 mL bottles, and stored in a Styrofoam box at 4 °C during collection. Upon arrival at the lab, the samples were centrifuged at 2700× *g* for 20 min at 4 °C and stored at −20 °C for later analysis [33].

### 2.8. Determination of Chemical Composition in Sow Colostrum and Milk

The colostrum and mature milk composition parameters included fat, solids-non-fat (SNF), density, protein, lactose, salt and pH using Master Eco, Milkotester LTD, Belovo, Bulgaria. Colostrum immunoglobulin G (IgG) was measured as Brix percent using 0.3 mL of colostrum sample and a digital Brix refractometer. This device is designed to measure the percentage of sucrose in liquids, approximates the total solids (TS%) content when used with non-sucrose-containing liquids. A commercial digital refractometer with a range of 0 to 55% Brix was used for measurements, specifically the Digital Brix Meter Sugar Refractometer from Deltatrak, CA, USA [34].

### 2.9. Analysis of Sow Colostrum Immunoglobulin

The concentrations of immunoglobulin G (IgG), immunoglobulin A (IgA), and immunoglobulin M (IgM) in sow colostrum was quantified using commercially available immunoglobulin-specific enzyme-linked immunosorbent assay (ELISA) kits (IgG, IgM, IgA quantitation kit; Sanwei Biological Engineering Co., Ltd., Shandong, China). The ELISA technique allows for the precise measurement of these immunoglobulins by utilizing specific antibodies that bind to the target proteins in the samples [35]. An automatic biochemical analyzer (RA-1000, Bayer Corp., Tarrytown, NY, USA) was employed for the quantification process. This automated system enhances the reproducibility of the results by standardizing the reaction conditions and ensuring consistency across all sample analyses. This method was critical for assessing the immune properties of colostrum as immunoglobulins such as IgG, IgA, and IgM are key components in passive immunity transfer from the sow to her piglets. Their concentrations provide insight into both maternal immune function and the effectiveness of colostrum in conferring immunity to newborn piglets [36].

### 2.10. Statistical Analysis

Descriptive statistics (i.e., means and standard error of the means) were calculated using the statistical package SAS 9.4 (SAS Institute, Cary, NC, USA). Sow performance was analyzed using multiple analyses of variance via the general linear model (GLM) procedure in SAS. Diarrhea scores were analyzed using the general linear mixed model procedure in SAS. The effects of treatment groups were analyzed using the general linear model procedure, and least-squares means were obtained for each parity class (3–4). Statistical significance was determined at *p* < 0.05. The statistical model formula for this study can be expressed as follows, tailored for the experimental design described:*Y_ijk_* = *μ* + *T_i_* + *P_j_* + *B_k_* + *ϵ_ijk_*
where *Y_ijk_* is the response variable (e.g., sow performance metrics, colostrum yield, piglet birth weight), μ is the overall mean of the response variable, *T_i_* is the fixed effect of the treatment (*i* = 1, 2, 3, 4, 5 for control and four supplement groups), *P_j_* is the fixed effect of parity (*j* = 3, 4 for sow parity classes), *B_k_* is the fixed block effect (e.g., initial body condition score or backfat thickness), *ϵ_ijk_* is the random error associated with each observation, assumed to be independently and normally distributed (*ϵ*~*N*(0,σ2)).

## 3. Results

### 3.1. Sow Reproductive Performance

The effects of miMCFA, lignocellulose, and HKL137 supplementation on sow reproductive performance are presented in Table 2. The live-born piglets and yield of colostrum production in the S1 and S4 group was significantly greater than those of the CON group. Conversely, no notable differences (*p* > 0.05) were observed in the duration of gestation, length of parturition, length of parturition per piglet, total feed intake, average daily feed intake, body weight, backfat thickness, total born piglets, alive percentage, piglet birth weight, stillborn piglets, mummified piglets, or colostrum intake per piglet.

### 3.2. Major Chemical Composition of Sow Colostrum and Mature Milk

Effect of miMCFA, lignocellulose, and HKL137 supplementation on the nutritional composition of sow colostrum and mature milk are displayed in Figure 1 and Figure 2, respectively. Fat content in colostrum was significantly increased in the S3 and S4 group (*p* < 0.01) than those of CON group. Additionally, IgG levels, measured using the Brix percentage, were significantly greater in the S3 and S4 groups compared to CON groups (*p* < 0.05). However, there were no differences in terms of protein, lactose, density, and solids-non-fat content in colostrum. By day 7 of lactation, the mature milk from all groups exhibited no significant variations in fat, protein, lactose, density, IgG levels, or solids-non-fat content.

### 3.3. Changes in Immunoglobulins of Sow Colostrum

The effect of miMCFA, lignocellulose, and HKL137 supplementation on the immunoglobulins alteration of sow colostrum is shown in Figure 3. The S3 and S4 groups demonstrated markedly elevated IgG levels compared to the CON group and other supplementation groups (*p* < 0.05). Likewise, IgM levels were notably higher in the S2 and S4 groups compared to the CON group and other treatments (*p* < 0.05). In contrast, IgA levels in sow colostrum showed no significant differences across the groups (*p* > 0.05).

## 4. Discussion

The results of the present study suggest that supplementing lactating sows with a combination of miMCFA, lignocellulose, and HKL137 (S4 group) can positively impact sow performance and colostrum composition. Specifically, the supplementation of miMCFA + Lig + HKL137 enhanced the amount of colostrum yield. Furthermore, the colostrum from supplemented sows had greater levels of beneficial fatty acids and immune-modulating compounds, potentially contributing to improved piglet growth and health. Previous studies suggest that dietary supplementation with medium-chain fatty acids (MCFA) positively affects sow performance and colostrum composition. According to Chen et al. [37], MCFA supplementation shortened the weaning-to-estrus interval of sows and increased the fat and protein content in colostrum. Lan and Kim [38] also observed improved performance in sows and their piglets with MCFA supplementation, including increased Lactobacillus counts and decreased *Escherichia coli* counts in fecal samples. Additionally, Jin et al. [39] reported that supplementation with various fat sources, including MCFA, improved growth performance of nursing piglets and increased the concentration of specific fatty acids in colostrum and milk.

Previous studies indicated that dietary supplementation with MCFA positively affects sow performance and colostrum composition. The benefits of fatty acids on the overall growth performance of piglets nursing from the sow are well documented. Supplementing sow’s diet with fatty acids during the latter stages of pregnancy and lactation increased piglet’s daily average body weight gain after weaning [40,41,42]. Previous research indicated that supplementing sows with 10% MCTs resulted in increased average daily body weight gain in nursing piglets. Feed intake and milk composition were identified as the primary parameters affecting growth performance [6,43,44]. Thus, the greater proportion of live-born piglets could be attributable to changes in fatty acid or calorie intake in addition to changes in sow milk composition due to dietary supplementation with the miMCFA + Lig + HKL137. The elevated levels of miMCFA in this study did not significantly impact the average daily feed consumption of sows. This study had a positive impact on sow backfat from late gestation through 7 days of lactation.

The supplementation of miMCFA + Lig + HKL137 improved the number of live piglets. These findings suggest that incorporating miMCFA, lignocellulose, and HKL137 into sow diets positively influences piglet survival. These results are consistent with previous research highlighting the potential of dietary supplements and additives to enhance piglet growth and health [45,46]. Previous studies suggested that dietary fiber supplementation in sow diets can have various effects on sow performance and colostrum composition. Yang et al. [47] reported that high-fiber diets increased the number of piglets born alive and improved litter weight. observed that high-fiber diets improved weaning weight and colostrum immunoglobulin levels. Gao et al. [48] found that different fiber sources had varying effects on farrowing performance and colostrum dry matter concentration. Feyera et al. [49] reported that high-fiber diets increased colostrum lipid content and colostrum intake in low-birth-weight piglets.

Fiber can play a role in volatile fatty acid (VFAs) production in the digestive system of sows. Fibers are complex carbohydrates that are not fully digestible in the small intestine of monogastric animals like pigs. Instead, they undergo fermentation in the hindgut (cecum and colon) by the microbial population present in the gastrointestinal tract [50,51]. During fermentation, microbes break down complex carbohydrates such as fiber into simpler compounds including VFAs. The primary VFAs produced in this process are acetic acid, propionic acid, and butyric acid. These VFAs serve as an energy source for the pig and can be absorbed and utilized [52]. The relationship between VFAs and milk fat production is not straightforward. While VFAs contribute to the overall energy supply for sows, the actual synthesis of milk fat occurs in the mammary glands. Triglycerides, the main components of milk fat, are produced within mammary cells from fatty acids obtained from dietary fats and mobilized body fat reserves. Despite fiber potentially contributing to VFA production and energy for sows, their direct influence on milk fat synthesis remains unclear [53,54]. However, supplementing sow diets with lignocellulose has been shown to improve gut health and an increase in the number of goblet cells in piglets, offering several benefits, including improved gut health.

Supplementation with miMCFA + Lig + HKL137 also demonstrated potential for enhancing piglet immune response. Additionally, HKL137 has shown promise in boosting the immune components of sow colostrum and milk. These findings suggest that supplementation with HKL137 in sow diets can potentially enhance the immune factors in both colostrum and milk. A previous study indicated that the effects of heat-killed *Lactobacillus plantarum* on sow performance and colostrum composition were influenced by various factors. Nardone et al. [55] reported that heat stress in cows during late pregnancy and the early postpartum period resulted in colostrum with lower concentrations of immunoglobulins and other components. Betancur et al. [56] highlighted that factors such as parity, genotype, endocrine status, and nutrition can affect colostrum yield and composition in sows, and demonstrated that oral administration of *Lactobacillus plantarum* CAM6 to breeding sows improved the content of lactose, nonfat solids, mineral salts, and the density of sow’s milk, while reducing milk fat. Additionally, Shang et al. [57] reported that fermented liquid feeding in sows resulted in lower coliform populations in feces and higher levels of lactic acid bacteria, which may contribute to improved colostrum quality. These findings suggest that heat-killed *Lactobacillus plantarum* and dietary factors can influence sow performance and colostrum composition.

Overall, the observed results suggest that the interactive mechanisms among miMCFA, lignocellulose, and HKL137 supplementation likely contribute to the enhanced performance of sows and improved colostrum composition. The miMCFA may play a role in providing a readily available energy source and modulating lipid metabolism, while lignocellulose could influence gut health through enhanced fiber fermentation and VFAs production. In addition, HKL137 may further enhance immune function as a heat-killed probiotic by modulating the immune components of colostrum and milk. These interactions collectively are likely to account for the improved sow productivity and piglet health. However, further investigations into these synergistic effects and underlying mechanisms are warranted to fully elucidate the observed outcomes.

## 5. Conclusions

The administration of miMCFA demonstrated the significant enhancement of live-born piglet numbers and elevated colostrum yield. Furthermore, the synergistic application of miMCFA in conjunction with lignocellulose and HKL137 exhibited additional neonatal piglet benefits and sow performance, specifically in terms of fat content and immunoglobulin G and M (IgG and IgM) concentrations in colostrum. These findings suggest potential mechanisms for enhanced maternal–offspring immune transfer, which may consequently lead to improved maternal health outcomes, superior colostrum quality, and enhanced immunological competence and growth performance in piglets. Finally, these results suggest that supplementing sow diets with miMCFA, lignocellulose, and HKL137 can significantly improve piglet survival and growth, offering a practical strategy for swine producers.

## Figures and Tables

**Figure 1 vetsci-12-00134-f001:**
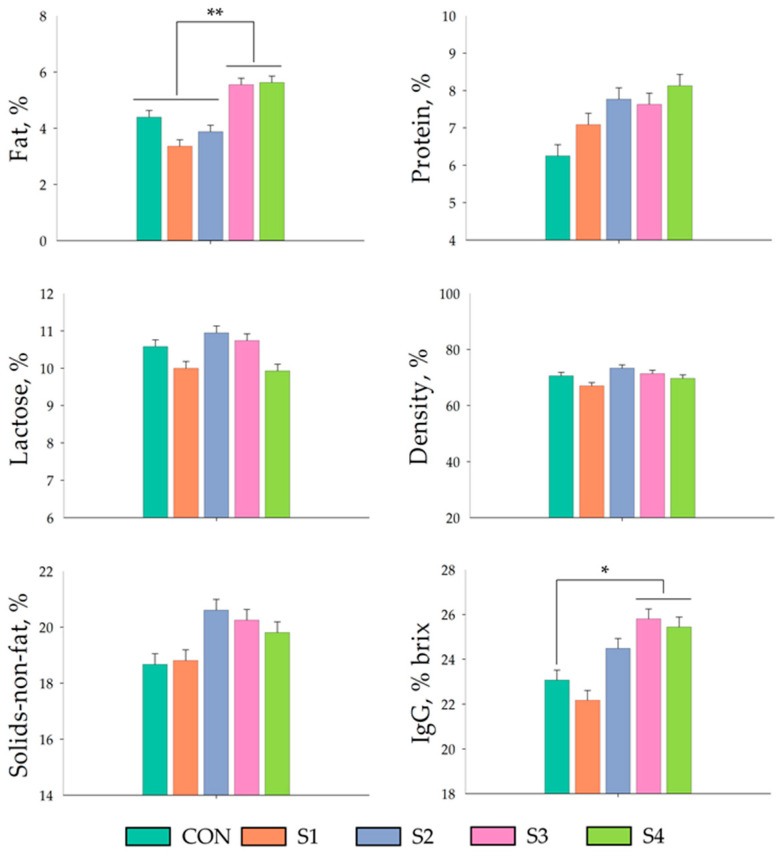
Effect of miMCFA, lignocellulose, and HKL137 supplementation on the nutritional composition of sow colostrum. ** = *p* < 0.01, * = *p* < 0.05.

**Figure 2 vetsci-12-00134-f002:**
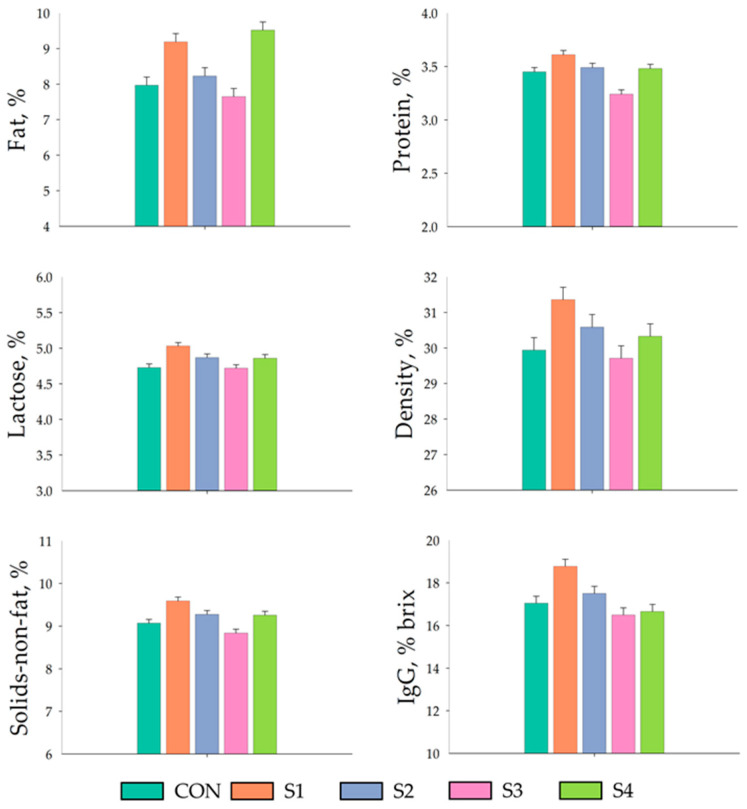
Effect of miMCFA, lignocellulose, and HKL137 supplementation on the nutritional composition of mature milk.

**Figure 3 vetsci-12-00134-f003:**
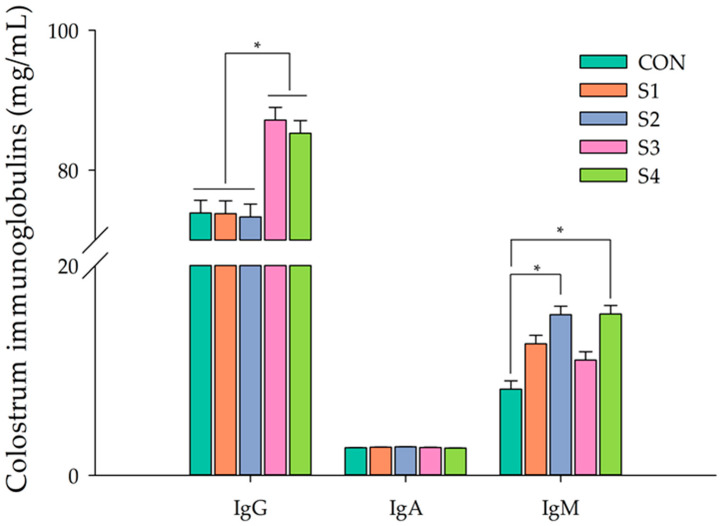
Effect of miMCFA, lignocellulose, and HKL137 supplementation on the immunoglobulin alteration of sow colostrum. * = *p* < 0.05.

**Table 1 vetsci-12-00134-t001:** The nutritional content of the basal diets analyzed during the gestation and lactation period.

Nutritional Content	Gestation Diet	Lactation Diet
Crude protein, %	14.56	18.76
Crude fat, %	4.95	7.53
Crude fiber, %	6.62	4.72
Ash, %	6.46	6.55
Moisture, %	12.37	11.42
Lysine, %	1.61	2.74
Metabolizable energy ^1^, kcal/kg	3394.31	3467.75

^1^ The value for each diet was calculated.

**Table 2 vetsci-12-00134-t002:** Effect of miMCFA, lignocellulose, and HKL137 supplementation on sow reproductive performance.

Item	CON	Dietary Supplement Treatment				SEM	*p*-Value
		S1	S2	S3	S4		
	*n* = 10	*n* = 10	*n* = 10	*n* = 10	*n* = 10		
Duration of gestation, day	112.6	112.4	112.5	113	112.8	0.12	0.536
Length of parturition, min	465.7	357.6	381.0	364.8	447.7	27.75	0.647
Length of parturition per piglet, min	42.8	23.7	27.9	31.9	32.0	2.86	0.294
Total feed intake							
Gestation day 100–114, kg	49.8	48.9	50.1	52.1	51.6	0.40	0.051
Lactation day 1–7, kg	23.1	23.7	24.8	23.5	26.4	0.89	0.792
Average daily feed intake							
Gestation day 100–114, kg	3.7	3.7	3.8	3.7	3.7	0.03	0.842
Lactation day 1–7, kg	2.9	3.0	3.1	2.9	3.3	0.11	0.792
Body weight							
Late gestation day 100, kg	226.0	233.2	230.5	220.7	228.7	4.44	0.938
Lactation day 7, kg	208.4	216.4	216.7	197.8	209.3	3.61	0.542
BW changes, kg	−17.6	−16.8	−13.8	−22.9	−19.4	3.44	0.963
BW changes, %	−7.2	−6.8	−4.1	−10.2	−8.4	1.47	0.829
Backfat thickness (P2)							
Late gestation day 100, mm	21.0	21.8	19.7	18.8	18.4	0.55	0.202
Lactation day 7, mm	18.6	21.2	19.3	18.8	18.8	0.54	0.460
BF changes, mm	−2.4	−0.6	−0.4	0.00	0.4	0.46	0.395
BF changes, %	−10.2	−2.3	−1.9	1.9	2.5	2.11	0.349
Total born piglets, n	14.7	17.6	15.7	14.6	16.1	0.51	0.335
Live-born piglets, n	11.5 ^b^	15.3 ^a^	13.9 ^ab^	12.8 ^ab^	14.4 ^a^	0.41	0.026
Alive percentage, %	79.7	87.2	88.6	80.7	90.0	1.62	0.147
Piglet birth weight, kg	1.4	1.4	1.5	1.5	1.4	0.03	0.679
Stillborn piglets, %	2.6	2.2	1.3	2.2	1.40	0.34	0.727
Mummified piglet, %	1.4	0.9	1.0	1.0	0.70	0.11	0.358
Colostrum production, kg/sow	4.3 ^c^	6.5 ^ab^	5.4 ^bc^	4.5 ^c^	6.9 ^a^	0.26	0.001
Colostrum intake per piglet, g/d/piglet	389.2	435.4	390.8	411.1	480.5	13.66	0.185

Data are expressed as means with the standard errors of the means (n = 10). CON, animals did not receive the supplement; S1, animals daily received 50 g microencapsulated medium-chain fatty acid in the morning meal as a topping; S2, animals daily received 50 g microencapsulated medium-chain fatty acid + 30 g lignocellulose in the morning meal as a topping; S3, animals daily received 50 g microencapsulated medium-chain fatty acid + 0.10 g heat-killed *Lactobacillus plantarum* strain L137 in the morning meal as a topping; S4, animals daily received 50 g microencapsulated medium-chain fatty acid + 30 g lignocellulose + 0.10 g heat-killed *Lactobacillus plantarum* strain L137 in the morning meal as a topping. Means with different superscripts within each row are significantly different (*p* < 0.05). SEM = standard errors of means.

## Data Availability

Data produced in this study are available from the corresponding. authors on reasonable request.

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
