# Peer review of "Effect of Microencapsulated Medium-Chain Fatty Acids, Lignocellulose, and Heat-Killed Lactobacillus plantarum L-137 Supplementation on Lactating Sow Performance, and Nutritional and Immunological Parameters in Colostrum"

_vetsci, 2025, doi:10.3390/vetsci12020134_

Round 1

Reviewer 1 Report

Comments and Suggestions for Authors

Dear Authors,

I have reviewed the paper entitled "Effect of microencapsulated medium-chain fatty acids, lignocellulose, and heat-killed Lactobacillus plantarum L-137 supplementation on lactating sow performance, and nutritional and immunological parameters in colostrum". Here are some suggestions for improvement: (Minor revision)

Introduction

1. Clarity and Conciseness: Simplify some long sentences to improve understanding.

o Example: “Feeding and nutrition of sows during late-gestation and lactation are crucial for maximizing piglet production and survival.” could be simplified to “Feeding and nutrition of sows during late-gestation and lactation are crucial for maximizing piglet production and survival.”

Summary

2. Key Results: Make sure the most important results are clearly highlighted.

o Example: “These findings suggest potential mechanisms for enhanced maternal-offspring immune transfer…” could be more specific about quantitative results.

Methodology

3. Specific Details: Provide more details about the methods used for data collection and analysis.

ie: Describe in more detail the microencapsulation process and how the supplements were administered.

Results

4. Please include all the information in each table to be consistent

Discussion

5. Comparison with Previous Studies: Expand the comparison with previous studies to better contextualize the findings.

ie: "Previous studies have shown similar improvements in colostrum quality with fatty acid supplementation…"

Conclusion

6. Practical Implications: More clearly highlight the practical implications of the findings for pig producers.

ie: "These results suggest that supplementing sow diets with miMCFA, lignocellulose, and HKL137 can significantly improve piglet survival and growth, offering a practical strategy for swine producers."

Best Regards

Author Response

Reviewer 1

I have reviewed the paper entitled "Effect of microencapsulated medium-chain fatty acids, lignocellulose, and heat-killed Lactobacillus plantarum L-137 supplementation on lactating sow performance, and nutritional and immunological parameters in colostrum". Here are some suggestions for improvement: (Minor revision)

Comment 1: (Introduction) Clarity and Conciseness: Simplify some long sentences to improve understanding. Example: “Feeding and nutrition of sows during late-gestation and lactation are crucial for maximizing piglet production and survival.” could be simplified to “Feeding and nutrition of sows during late-gestation and lactation are crucial for maximizing piglet production and survival.”

Response 1: We agree with this comment. Therefore, we simplified some long sentences according to your suggestion (Line 49-50, highlighted in yellow).

Comment 2: (Summary) Key Results: Make sure the most important results are clearly highlighted. Example: “These findings suggest potential mechanisms for enhanced maternal-offspring immune transfer…” could be more specific about quantitative results.

Response 2: Thanks for making that point. We've updated those lines (27-31, highlighted in yellow) based on your feedback as follow: “Compared to the control group, sows with enhanced immune transfer produced 60.5% more colostrum during their first piglet parturition period. The colostrum exhibited increased concentrations of lipids (28%) along with significant IgG and IgM levels. These results highlight our strategic feeding leads to higher-quality colostrum, which results in higher number of live-born piglets and birth weights.”

Methodology

Comment 3: Specific Details: Provide more details about the methods used for data collection and analysis. ie: Describe in more detail the microencapsulation process and how the supplements were administered.

Response 3: We appreciate the reviewer’s comment.  Detail of microencapsulation process was provided in line 117-121 (highlighted in yellow). In addition, each supplemental group was administered daily in the morning meal as a topping (Line 146-147, highlighted in yellow).

Results

Comment 4: Please include all the information in each table to be consistent

Response 4: Thanks very much for pointing that out. After double-checking all the numbers in every table, we found some mistakes, so we revised them (Table 2, highlighted in blue).

Discussion

Comment 5: Comparison with Previous Studies: Expand the comparison with previous studies to better contextualize the findings. ie: "Previous studies have shown similar improvements in colostrum quality with fatty acid supplementation…"

Response 5: We revised this section according to your suggestion (Line 58-60, highlighted in blue) as follows: “Previous studies have shown similar improvements in colostrum composition and piglet growth following supplementation with fatty acid supplementation including β-hydroxy β-methyl butyrate, polyunsaturated fatty acids, and medium-chain fatty acids [5-7].” However, this paragraph has been moved to the introduction section following Reviewer 2.

Conclusion

Comment 6: Practical Implications: More clearly highlight the practical implications of the findings for pig producers. ie: "These results suggest that supplementing sow diets with miMCFA, lignocellulose, and HKL137 can significantly improve piglet survival and growth, offering a practical strategy for swine producers."

Response 6: We revised this section according to your suggestion (Line 377-380, highlighted in yellow).

Reviewer 2 Report

Comments and Suggestions for Authors

The manuscript investigates how microencapsulated medium-chain fatty acids (miMCFA), lignocellulose, and L137 affect the performance of sows and their offspring. While the study is significant to pig production, several issues must be addressed to meet the publishable standards.

Firstly, the ingredients of basal diets for late gestation and lactating sows should be presented in Table 1. Furthermore, it has been observed that the metabolizable energy (ME) content was below the recommended level of 3,300 kcal/kg for gestating sows (>90d), could you please clarify the methodology used to determine the ME content of the two basal diets? (Refer to Table 1 for the analyzed nutritional …)

Second, the data presented in Table 3 exhibits inconsistencies, for instance, the total feed intake during the 7-day lactation in the control group is reported as 3.7 kg, whereas the average daily feed intake (ADFI) for the same duration is indicated to be 2.9 kg. Furthermore, the ADFI is recorded as 23.1 kg at 14 days of gestation, with a total feed intake of 49.8 kg in the control group. Similar discrepancies are observed in the other groups as well. It is imperative to verify the data concerning for body weight (BW) and body fat (BF) changes to ensure their accuracy.

Third, was it possible to compare the litter weights among different groups? After standardizing the litter size to 11 piglets, the litter weight in the Control group (16.7 kg) was lower than that of the other treatments (19.5, 21.8, 19.1, and 19.9 kg). Therefore, were the weight gain and ADG in the litter during 0-3 days or 0-7 days rendered meaningless? Please check and address this in detail.

Finally, the results should be further discussed regarding the interactive mechanisms among miMCFA, lignocellulose, and L137 supplementation. Additionally, an exploration of the underlying factors contributing to the observed performance difference among sows and piglets across the various groups. Additionally, incorporating the first paragraph of the Discussion section into the Introduction is suggested.

Author Response

Reviewer 2

The manuscript investigates how microencapsulated medium-chain fatty acids (miMCFA), lignocellulose, and L137 affect the performance of sows and their offspring. While the study is significant to pig production, several issues must be addressed to meet the publishable standards.

Comment 1: Firstly, the ingredients of basal diets for late gestation and lactating sows should be presented in Table 1. Furthermore, it has been observed that the metabolizable energy (ME) content was below the recommended level of 3,300 kcal/kg for gestating sows (>90d), could you please clarify the methodology used to determine the ME content of the two basal diets? (Refer to Table 1 for the analyzed nutritional …)

Response 1: Thank you for your helpful feedback and attention to detail. After reviewing our original data, we discovered some errors and have corrected the values in Table 1 (highlighted in blue).

Comment 2: Second, the data presented in Table 3 exhibits inconsistencies, for instance, the total feed intake during the 7-day lactation in the control group is reported as 3.7 kg, whereas the average daily feed intake (ADFI) for the same duration is indicated to be 2.9 kg. Furthermore, the ADFI is recorded as 23.1 kg at 14 days of gestation, with a total feed intake of 49.8 kg in the control group. Similar discrepancies are observed in the other groups as well. It is imperative to verify the data concerning for body weight (BW) and body fat (BF) changes to ensure their accuracy.

Response 2: Thank you very much for your helpful suggestion and observation. After reviewing the original data, we identified some errors and have accordingly revised certain values in Table 2 (highlighted in blue).

Comment 3: Third, was it possible to compare the litter weights among different groups? After standardizing the litter size to 11 piglets, the litter weight in the Control group (16.7 kg) was lower than that of the other treatments (19.5, 21.8, 19.1, and 19.9 kg). Therefore, were the weight gain and ADG in the litter during 0-3 days or 0-7 days rendered meaningless? Please check and address this in detail.

Response 3: Thanks very much for pointing that out. Following your suggestion, I have removed Table 3 and the result section "3.2 The Litter Parameters" as they were not meaningful. However, all data regarding to “Litter weight parameters“ has been deleted in the whole manuscript.

Comment 4: Finally, the results should be further discussed regarding the interactive mechanisms among miMCFA, lignocellulose, and L137 supplementation. Additionally, an exploration of the underlying factors contributing to the observed performance difference among sows and piglets across the various groups. Additionally, incorporating the first paragraph of the Discussion section into the Introduction is suggested.

Response 4: Thank you for the reviewer’s comment. I have provided additional discussion in response to the reviewer’s remark (Line 359-368, highlighted in blue). Anyway, first paragraph of the Discussion section was moved into the Introduction (Line 58-60, 82-85 and 101-103, highlighted in blue). All references numbers in the entire manuscript have been reorganized (highlighted in green).

Reviewer 3 Report

Comments and Suggestions for Authors

Dear Authors, your work is very interesting and important, highlighting the importance of functional nutrients in sow and piglet's performance, and health. 

I have some remarks, as follows:

Please provide keywords that do not appear in the title.

Please provide the hypotheses of the work right after the objectives.

Please provide the entire statistical model(s) and how the data were handled before ANOVA. Were the data normal? Provide also the comparison test used.

On the footnote of each table, please, include the superscript letters hierarchy (a>b>c?) and the comparison test.

Please correct all the references following the guidelines. Remember that they must be complete and scientifically relevant.

Best regards,

Author Response

Reviewer 3

Dear Authors, your work is very interesting and important, highlighting the importance of functional nutrients in sow and piglet's performance, and health. I have some remarks, as follows:

Comment 1: Please provide keywords that do not appear in the title.

Response 1: Thanks very much for pointing that out. “colostrum quality”was added as a new keyword (Line 46, highlighted in green).

Comment 2: Please provide the hypotheses of the work right after the objectives.

Response 2: Thank you for the reviewer’s comment. I have provided hypotheses of the work (Line 112-114, highlighted in green).

Comment 3: Please provide the entire statistical model(s) and how the data were handled before ANOVA. Were the data normal? Provide also the comparison test used.

Response 3: We revised this section according to your suggestion (Line 230-238, highlighted in green).

Comment 4: On the footnote of each table, please, include the superscript letters hierarchy (a>b>c?) and the comparison test.

Response 4: Thank you for your helpful attention to detail. I have provided the footnote of each table (Table 2, highlighted in green).

Comment 5: Please correct all the references following the guidelines. Remember that they must be complete and scientifically relevant.

Response 5: Thanks very much for pointing that out. After double-checking, all references were prepared according to the guidelines.

Round 2

Reviewer 2 Report

Comments and Suggestions for Authors

The manuscript has shown significant improvement. However, it should be noted that metabolizable energy should be calculated rather than analyzed. Please make the necessary corrections in Table 1.

Author Response

Comment 1: The manuscript has shown significant improvement. However, it should be noted that metabolizable energy should be calculated rather than analyzed. Please make the necessary corrections in Table 1.

Response 1: Metabolizable energy values for gestation and lactation diets were reviewed and updated. Then, the footnote was added below Table 1 as follows: “The value for each diet was calculated.” (Highlight in yellow).